# Application of the Yamanaka Transcription Factors *Oct4*, *Sox2*, *Klf4*, and *c-Myc* from the Laboratory to the Clinic

**DOI:** 10.3390/genes14091697

**Published:** 2023-08-26

**Authors:** Marisol Aguirre, Manuela Escobar, Sebastián Forero Amézquita, David Cubillos, Camilo Rincón, Paula Vanegas, María Paula Tarazona, Sofía Atuesta Escobar, Juan Camilo Blanco, Luis Gustavo Celis

**Affiliations:** 1Department of Genetics, Fundación Valle del Lili, Cali 760026, Colombia; maguirre11@ksu.edu; 2Faculty of Medicine, Universidad Icesi, Cali 760031, Colombia; 3Faculty of Medicine, Universidad de La Sabana, Km 7, Autopista Norte de Bogotá, Chía 250001, Colombia

**Keywords:** cell- and tissue-based therapy, *Oct4*, *Sox2*, *Klf4*, *c-Myc*, cancer, neurodegenerative diseases, induced pluripotent stem cells, rejuvenation

## Abstract

The transcription factors *Oct4*, *Sox2*, *Klf4*, and *c-Myc* enable the reprogramming of somatic cells into induced pluripotent cells. Reprogramming generates newly differentiated cells for potential therapies in cancer, neurodegenerative diseases, and rejuvenation processes. In cancer therapies, these transcription factors lead to a reduction in the size and aggressiveness of certain tumors, such as sarcomas, and in neurodegenerative diseases, they enable the production of dopaminergic cells in Parkinson’s disease, the replacement of affected neuronal cells in olivopontocerebellar atrophy, and the regeneration of the optic nerve. However, there are limitations, such as an increased risk of cancer development when using *Klf4* and *c-Myc* and the occurrence of abnormal dyskinesias in the medium term, possibly generated by the uncontrolled growth of differentiated dopaminergic cells and the impairment of the survival of the new cells. Therefore, the Yamanaka transcription factors have shown therapeutic potential through cell reprogramming for some carcinomas, neurodegenerative diseases, and rejuvenation. However, the limitations found in the studies require further investigation before the use of these transcription factors in humans.

## 1. Introduction

Induced pluripotent cells (iPSCs) are reprogrammed somatic cells that can differentiate into every cell type of the three germ layers, the endoderm, the ectoderm, and the mesoderm. They have the potential of unlimited proliferation and differentiation, like embryonic stem cells (ESs), and give rise to different cells and tissue types, such as connective, epithelial, muscle, and nervous cells and tissues. Fibroblasts are the first and most frequent somatic cell type used for iPSCs, and are obtained from a skin biopsy, an invasive procedure, but keratinocyte, urine, and blood cells could also be used with less invasive techniques [1]. Yamanaka and Takahashi discovered the core transcriptional factors required for reprogramming somatic cells to induce pluripotent stem cells, *Oct4*, *Sox2*, *Klf4*, and *c-Myc* (OSKM) [2], which became the milestone for designing studies on various diseases, pharmacological trials, the design of new drugs, and both cell and regenerative therapy. A transfer system is required to insert the OSKM genes into somatic cells to generate the iPSCs, which can be integrative or nonintegrative [3]. Viruses such as retroviruses or lentiviruses, plasmids, and transposons, which are mobile genetic elements used to deliver exogenous pluripotency genes, are used for an integrative system. Methods for a nonintegrative system that do not insert exogenous DNA into the host genome include the use of adenoviruses, cytoplasmic RNA, episomal vectors, and polycistronic minicircle DNA nonviral vectors [3]. A nonintegrative nonviral transfer system can be easy to use, but less efficient than a lentiviral vector [3]. When the reprogramming transcription factors are in the somatic cell, they bind to pluripotency-associated recognition sequences in somatic cells to start chromatin changes due to DNA methylation, histone modifications, and ATP-dependent chromatin remodeling [3]. *c-Myc* opens chromatin by binding to a methylated region. Then, *Oct4* and *Sox2* interact with the enhancers and promoters of genes directly involved in the identity of the somatic cell and other reprogramming genes. Mesenchymal gene expression is silenced [3]. OSKM components also bind to each other to create a self-regulating interconnected loop that triggers their promoter and the enhancers and promoters of other genes involved in development, growth, differentiation, and the early pluripotency stage. Then, the somatic cells change their morphology, undergo a mesenchymal-to-epithelial transition, and proliferate [3]. During the transition, the expression of pluripotency genes is activated, and epithelial genes such as *Cdh1*, *Epcam*, and *Ocln* up-regulate to create larger ES-like clusters [3].

*Oct4* specifically expresses all pluripotent cells during embryogenesis and undifferentiated embryonic stem cells. It is required to establish and maintain cellular pluripotency. Studies have found that the lack of this transcription factor stops the development of the internal cell mass of the blastocyst and causes a high embryo mortality [4], but over-expression causes cellular differentiation induction [5]. *Sox2* is important for the embryonic development of tissues and organs. *Sox2* interacts with *Oct4* to regulate the expression of genes such as *Nanog*, *Fgf4*, *osteopontin*, and *lefty*, which are involved in maintaining cellular pluripotency [6]. *Sox2* is expressed not only in pluripotent cells, but also in the late phases of embryo development, especially in neural stem cells [6]. *Klf4* contains three tandem zinc fingers to interact directly with *Oct4* and is expressed at sufficient levels to generate iPSCs. *Nanog* requires *Klf4* for its activation [7]. The forced expression of *Oct4*, *Sox2*, *Klf4*, and *Nanog* can revert fibroblasts to a pluripotent state, especially *Nanog*, a homeodomain transcription factor crucial in the earliest steps of embryogenesis [7]. *Nanog* interacts with *Oct4*, *Sox2*, the nucleosome remodeling and deacetylase (NuRD) complex, and polycomb repressive complex 2 to form homo- and heterodimers necessary for ES self-renewal and pluripotency by regulating gene expression [8]. A high expression of *Nanog* induces cellular self-renewal, but a low expression causes differentiation [9]. Nucleosome position is related to gene expression. Its depletion and histone tails covalent modifications are necessary for opening chromatin and the later binding of transcription factors that allow obtaining iPSCs. Thus, nucleosome position is part of the reprogramming process [10]. *c-Myc* has a role in inducing apoptosis and cell differentiation, growth, and metabolism [9,11]. It has multiple domains and heterodimerizes with Max, a cofactor for *c-Myc* that helps its transcriptional activation and repression by forming a heteromeric complex [9,11]. It is also an oncogene that participates in processes of cell proliferation and inhibits differentiation, DNA replication, and metastasis [9]; mice transplanted with iPSCs induced by *c-Myc* developed teratomas [12]. *c-Myc* is involved in self-renewal processes and induces epigenetic changes that cause dedifferentiation or block cell differentiation [3,13,14].

Finally, reprogramming somatic cells to induce pluripotent stem cells creates a source of stem cells for the development of therapies for cancer, neurodegenerative diseases, and regeneration, such that we considered reviewing the clinical applications of Yamanaka transcription factors in the literature.

## 2. Applications of the Yamanaka Transcription Factors in Cancer

A literature review found several applications of OSKM transcription factors in cancer-related experiments [15]. Technologies based on cell regeneration and reprogramming allow cancer stem cells to be reprogrammed into pluripotent stem cells using the described transcription factors [15]. However, this may be controversial, since *c-Myc* and *Klf4* factors increase the risk of tumorigenesis [15]. *Oct4* overexpression was present in multiple cancers, such as ovarian, cervical, colorectal, liver, breast, and bladder cancers. *Sox2* was present in at least 25 types of cancers, including breast, gastric, and pancreatic cancers; adenocarcinoma of the ampulla of Vater; malignant gliomas; and other brain tumors. *Klf4* overexpression was found in squamous cells and other cancer-forming processes [16], and *c-Myc* was the main oncogenic factor related to tumorigenesis and glycolysis as well as being linked to Wnt/β-catenin signaling in lung cancer [17].

Likewise, a study combining gene and cell therapy was performed in a mouse model with anemia to recognize the therapeutic potential of pluripotent stem cells by using mouse embryonic fibroblasts, but this is not possible in human treatments. However, human dermal fibroblasts have been reported, with potential clinical applications in various fields, for example, genetic diseases. The study aimed to compare the results obtained from using bone marrow stromal cells and embryonic fibroblasts. Mice aged 12 to 24 weeks were isolated, and mouse bone marrow mononuclear cells were collected using centrifugation and fused. Then, hybrid cell induction was performed, followed by the use of a transfer method involving retroviruses containing the Yamanaka transcription factors. The result was that embryonic fibroblasts did not work well for generating pluripotent cells, but after using a higher transduction efficiency, it was possible. Bone marrow mononuclear cells did not require such interventions. Subsequently, these were transplanted subcutaneously to test the pluripotency of the bone marrow cells. Four weeks after injection, encapsulated cystic tumor growth was observed in all mice, and microscopic examinations showed that these tumors contained various tissues, including neural tissue, epidermal tissue, muscle fiber cells, cartilage, and pancreas-like cells, and the differentiation of three germ layers was confirmed in vitro [16].

Yamanaka transcription factors show participation in some processes of tumorigenesis and pluripotent stem cells when interacting with other molecules that can offer new research options. For example, the helicase DNA-binding protein (CHD4) suppresses the expression of *Sox2* and regulates cancer stem cells, targeting different molecules such as SANIL1, CCND1, CCND3, P21, P7, and *c-Myc*. CHD4 is part of the nucleosome remodeling and deacetylase complex that affects the pluripotency and differentiation of embryonic stem cells. CHD4 expression is associated with developing glioblastomas and colorectal, hepatocellular, endometrial, and breast cancers. A high CHD4 expression is associated with a worse survival prognosis, a larger tumor size, resistance to drugs used in chemotherapy, and a lower migration rate without affecting tumor cell proliferation. A low CHD4 expression causes increasing levels of *Sox2* and paclitaxel resistance [18].

On the other hand, this technology could help to reduce the malignant potential of certain tumors such as sarcomas. In in vivo mice experiments, *Nanog* and *Lin28* were used in addition to the four Yamanaka factors in five sarcomatous cell lines, and these experiments showed a decrease in the tumor growth rate compared with the controls. A smaller size and a decrease in the number of tumor cells was observed in mice with reprogrammed cells compared with the control group. In addition, the experiments demonstrated that the reprogrammed sarcomas could differentiate into mature connective tissue and red blood cells, which is evidence that the use of transcription factors decreases tumor aggressiveness and changes the morphology [19].

In an experiment in which cancer stem cells were split from samples taken from the peripheral blood mononuclear cells of a patient with neuroendocrine carcinoma of the lung that was given episomal vectors to generate pluripotent stem cells, at a follow-up two weeks later, colonies with the typical morphology of embryonic stem cells, a good in vitro differentiation potential, and a high expression of pluripotent markers were detected, which made it possible to determine that a treatment could be carried out in the same place where the disease is found. However, it is also relevant to evaluate the safety and effectiveness of these therapies according to each tumor [20].

*Sox2*, *Oct4*, and *Nanog* also demonstrated a high diagnostic potential and a potential future use in targeted therapy for adenocarcinoma and squamous cell lung cancer. For example, a clinical study revealed that the mentioned transcription factors were present in 76% of the 30 lung tissue samples collected before the treatment. *Oct4* overexpression was associated with more advanced stages of lung cancer and tumors with little differentiation. Another study with 147 subjects showed a correlation of *Sox2* with squamous cell lung cancer, with *Sox2* being present in 79% of the samples, demonstrating its crucial role in the diagnostic approach to this pathology with a high incidence and mortality in the world today. *Sox2* in non-small cell lung cancer has demonstrated the ability to preserve the pluripotent capacity of these cells, generating resistance to anticancer therapy with paclitaxel. Thus, inhibiting *Sox2* and its effector CIC-3 increases the sensitivity of the drug to the affected cells, providing a new therapeutic option for this type of lung cancer [21].

Applications of the Yamanaka transcription factors in cancer play a role in treatment, diagnosis, prognosis, and drug resistance, but more research is needed on these factors to understand their safety, effectiveness, and interaction with other molecules.

## 3. Applications of Yamanaka Factors in Neurodegenerative Diseases

Yamanaka transcription factors have impacted the development of cell replacement therapies in neurodegenerative diseases due to their ability to reprogram somatic cells (Figure 1). A study with rats achieved differentiation into neural stem cells, neurons, and dopaminergic neurons with iPSCs from the OSKM transcription factors, generating a slight reduction in the classic motor impairments of Parkinson’s disease [22]. In another study, for olivopontocerebellar atrophy, iPSCs were developed from fibroblasts to obtain neural cells identical to those of the patient with a 99% accuracy, which may be key for the treatments of this neurodegenerative disease [23]. While there are positive impacts, unfortunately, there were also adverse effects in both studies mentioned above. A few cell colonies were reported, with the growth of teratomas in mice with a severe combined immunodeficiency [22,23].

These transcription factors show a potential application for the development of a possible treatment for Parkinson’s disease. In in vivo studies, it is possible to obtain pluripotent cells differentiated into dopaminergic cells that could treat this neurodegenerative disease [24]. Additionally, iPSCs transplanted into a mouse brain can migrate to various brain regions and differentiate into dopaminergic, GABAergic, and glutamatergic neuronal cells and glial cells that are functional, providing hope for a treatment for multiple diseases [24]. Moreover, transplanting stem cells converted into dopaminergic neurons in the striatum of mice with motor abnormalities generated a decrease in motor symptoms [24]. However, in the medium term, dyskinesia was evident in the transplanted mice, which raised doubts about the safety of this therapy for improving motor symptoms in Parkinson’s disease [24]. Additionally, the long-term cell survival, functionality, and efficacy are unknown.

IPSCs obtained from the peripheral blood mononuclear cells (PBMCs) of schizophrenia patients and healthy control patients showed that these cells presented higher and more severe inflammatory responses than normal iPSCs, indicating that inflammation and the immune system could play a role in the etiology of schizophrenia disorders [25]. An analysis of these data helped to understand the pathogenesis of some neurodegenerative diseases. In a study, induced neural stem cells (iNSCs) generated from mesenchymal stromal cells (MSCs) encoding OSKM from a cynomolgus monkey were transplanted into an immunodeficient model of Parkinson’s disease induced by 6-hydroxydopamine (6-OHDA), a neurotoxin used to generate the destruction of dopaminergic neurons in the nigrostriatal region and its consequent neurodegeneration [26]. They demonstrated a good plasticity that allowed them to differentiate into dopaminergic neurons and glial cells such as astrocytes, oligodendrocytes, and pan-neurons, resulting in effective results in the improvement of motor functions in comparison with a control group six months post-transplantation, with a high statistical significance [26]. This was probably due to the increase in dopamine production and the reduction in the loss of dopaminergic neurons, indicating a neuroprotective factor against the progressive degeneration caused by the disease.

Neurodegenerative diseases such as amyotrophic lateral sclerosis (ALS) generate the loss of lower and upper neurons, which produces progressive weakness, spasticity, muscle atrophy, and ultimately, death [27], and they have gained more importance and relevance since iPSCs obtained from ALS patients were found to differentiate into motor neurons [28]. Studies carried out with iPSCs, especially using the transcription factors *Oct4* and *Klf4*, have resulted in an increase in the preservation of spinal cord neurons at the injection site and an increase in the expression of neuronal growth factor (NGF). It has also been evidenced that iPSCs are neuroprotective, with neuromodulator effects, and that they can reduce disease progression [29]. Additionally, thanks to their implementation in this disease, the modeling of ALS with iPSC-derived neurons has elucidated their potential role in drug discovery [30]. In 2021, a study described the role of cell lines in the pathogenesis and progression of ALS and how to use these cells derived from human iPSCs to identify new therapeutic targets [31]. Another study described how to produce motor neurons from human iPSCs, their application in drug discovery for this disease, and their potential use in cell therapy by performing cell transplantation [32].

For frontotemporal dementia and Alzheimer’s disease modeling and drug discovery, studies show favorable results [33]. iPSC therapy improves neuronal plasticity and memory by increasing proteins responsible for cognition. For example, in a mouse model, iPSC therapy could achieve an increased release of brain-derived neurotrophic factors that improve cognitive function. iPSCs differentiate into different brain cells, promoting brain neuroplasticity, decreasing the production of inflammatory cytokines, and stimulating growth factor release. However, tumor growth could appear with this therapy, so more studies are still needed [34].

Likewise, recent studies have investigated and understood the pathophysiology of neurodegenerative diseases to provide timely treatment. According to these investigations, the excessive activation of microglia in the brain and increased levels of proinflammatory cytokines such as TNF-α, IL-1β, IL-6, and IL-10 were present. A novel protein called BIG1 (brefeldin A-inhibited guanine nucleotide-exchange protein 1), responsible for the cell migration of iNSCs and neuronal soma growth and formation, was also discovered. RT-qPCR and Western blot techniques found that silencing BIG1 reduced the expression of TNF-α, IL-1β, and IL-6, decreasing neuroinflammation, and the inhibition of this protein generated a decrease in the cell migration of iNSCs. Based on the above, the transcription factor *Klf4* binds to the BIG1 promoter, positively regulating it, thus mediating neuroinflammation and allowing cell migration through the PI3K/Akt/NF-kB signaling pathway that regulates proliferation, apoptosis, and differentiation, controlling a wide range of target proteins and, thus, improving neuroinflammation and neuroplasticity [35].

With the passage of time and further research, these OSKM transcription factors could be applied to improve the modeling of many more diseases and to gain a deeper understanding of disease pathogenesis, progression, and therapeutic targets.

## 4. Applications of Yamanaka Transcription Factors in Rejuvenation

In the years following the discovery of the Yamanaka transcription factors, efforts were made to implement a method of rejuvenating cells, tissues, and even organisms using them. Senescent cells can be reprogrammed in two ways: first, through developmental reprogramming using OSKM or somatic cell nuclear transfer (SCNT), and second, through age reprogramming by bypassing the de-/re-differentiation cycle, reducing or silencing age-related markers, and retaining their original identity. In the proposed model for age reprogramming, the old cell transiently de-differentiates after OSMK introduction. Then, a phase of epigenetic instability occurs in which aging-related markers are reduced or silenced. Later, OSKM expression stops, and ESs/iPSCs re-differentiate into an original young cell without embryonic features (Figure 2). The time that the cell spends in the phase of epigenetic instability can determine the degree of rejuvenation. An optimal time results in a younger reprogrammed cell without losing its somatic identity, but a very long time generates a reprogrammed cell that is less young. The difference in the results of chronological reprogramming may be due to the accumulation of senescence-associated gene products at the end of the critical reprogramming time [36]. However, more research is needed.

In 2010, the possibility of inducing cell regression without generating stem cells was proposed to avoid the risk of cancer, giving rise to partial cell reprogramming and epigenetic rejuvenation [37,38], which allowed its subsequent use in human fibroblasts to restore their heterochromatin to non-senescent levels [39] and in mouse fibroblasts to decrease age-related changes such as DNA damage, nuclear damage, stress, and senescence factors, among others [40,41,42].

In 2016, a study used OSKM to generate partial reprogramming in vivo in progeroid mice, extending their lifespan without the appearance of teratomas. When applied in adult mice, it improved the regenerative capacity of muscles and the pancreas after injury, as well as glucose tolerance [43]. However, this study did not have a method to quantify the degree of rejuvenation [44]. In another study, it was not possible to demonstrate the duration of rejuvenation over time at the cellular or organism level [45], which gave rise to subsequent studies aimed at improving the phenotype of mice with Hutchinson–Gilford progeria syndrome and promoting cell regeneration in middle-aged mice by administering OSKM and producing partial cellular reprogramming. However, these effects remained transient [46].

Thanks to previous discoveries, OSKM was used with *Lin28* and *Nanog* in other cell lines, such as mature neurons or neurons with aging lesions, to rejuvenate them through their total or partial reprogramming [47]. At the level of retinal ganglion cells (RGCs), the overexpression of at least three of the transcription factors, including *Lin28*, was performed using the AAV2 viral vector, resulting in the significant regeneration of the optic nerve and axons, as well as a decrease in injury-induced neuronal cell death. However, using these factors individually or using only two did not regenerate the cell [46]. Another significant point is that the experimental approach and the modified transcripts in mature neurons may have potential problems. For example, some of the transcription factors used are oncoproteins, such as *Sox2*, *Lin28*, and *c-Myc*, which may favor tumor creation when overexpressed in glial cells with a cell division capacity [47].

Additionally, the short-term expression of Yamanaka factors promotes tissue regeneration in vivo. For example, in myofibers, it favors muscle regeneration in young mice, inducing the activation of muscle stem cells or satellite cells (SCs), thus accelerating their regeneration. This occurs through the modification of the stem cell niche, as the regenerative capacity of SCs is influenced by intrinsic modulators and the extrinsic microenvironment. In this case, the expression of the factors in myofibers regulates the expression of genes for the SCs microenvironment, including the up-regulation of p21, which, in turn, down-regulates Wnt4. Myofibers secrete Wnt4 to maintain SCs quiescence, favor muscle regeneration, and slow SCs aging. In contrast, the expression of Yamanaka factors directly in SCs does not enhance muscle regeneration [48].

Finally, in this manuscript, two ongoing investigations are dealing with the administration in mice of *Oct4*, *Sox2*, and *Klf4* factors in combination through a lentivirus and doxycycline, with the latter being used to activate the factors administered in the lentivirus, which has resulted in a reversal of biomarkers of aging, such as a decreased frailty index and an increased epigenetic age calculated from methylation patterns, thereby prolonging the lifespan of the mice to 109% and maintaining it over time [49,50]. However, there are limitations, such as the difficulty with the combined administration of these transcription factors due to their size, which exceeds the capacity of lentiviruses; the formation of teratomas with the use of *c-Myc*; and the possibility of replicating these results in humans (Figure 3) [49]. Finally, the Yamanaka transcription factors have a high potential for use in the aging field. However, the mechanism and risks must be well known before these factors can be safely implemented as a treatment in humans [51].

## 5. Discussion

Induced pluripotent cells from reprogrammed somatic cells can differentiate into every cell type and possess the potential for unlimited proliferation and differentiation, like ESs, allowing for different cells and tissue types to be raised [1]. As has been mentioned, thanks to Yamanaka and Takahashi’s discovery in 2006 when they reprogrammed mouse fibroblasts to form iPSCs [52], a new panorama has been opened by being able to perform in-depth studies and clearly understand the pathogenesis of different diseases, in addition to proposing new therapeutic measures. Since the discovery of the mechanisms for cell reprogramming, they have undergone more development. The first attempt at cell reprogramming involved transferring genes to fibroblasts based on integrated retroviral vectors. Subsequently, different strategies were developed to avoid the gene integration of foreign DNA, including the non-integrated viral and non-viral transfer systems mentioned in our review [53,54,55,56,57,58]. In recent years, based on cell reprogramming performed in mouse cells, the possibility of chemically reprogramming human somatic cells has been raised, considering that these are difficult to stimulate chemically due to their stable epigenome [59,60,61] and their reduced plasticity [62,63], laying the foundations for regenerative therapies of human cells [64].

The transcription factors have been used in several in vivo and in vitro studies, in both mouse and human cells, producing very encouraging results such as a reduction in the size and aggressiveness of certain tumors such as sarcomas [19], the production of dopaminergic cells in the case of Parkinson’s disease [24], neuronal cells to replace those affected by olivopontocerebellar atrophy [23], the regeneration of the optic nerve due to trauma [47], and cellular, tissue, and organism rejuvenation [48,49,50].

However, there are several limitations. First, *Klf4* and *c-Myc* have an oncogenic potential that creates a dilemma regarding which factors are the most suitable for reprogramming cells, so the use of transcription factors is still under study [15,49,50]. Second, there is a bioethical issue because the transcription factors are not safe to use yet in humans. After all, abnormal dyskinesias were found in mice in the medium term, possibly generated by the uncontrolled growth of differentiated dopaminergic cells. It is necessary to be sure of the safety of this therapy before using it in a patient with Parkinson’s disease [24]. In addition, there are no standard guidelines for its use, since the Yamanaka transcription factors and other factors, such as *Lin28* and *Nanog*, have been combined with controversial results [19,47]. Third, there are factors such as the environment in which the transcription factors can interact with pros or cons to the results. The interaction of the cell, the host, and the disease can influence the efficiency of the reprogramming and would affect the survival of the new cells and their duration, as observed in the case of in vivo studies on Parkinson’s disease, where iPSCs differentiated into dopaminergic cells lost their effectiveness over time. The reason for this decline was that when they were found in the pathological environment of this condition, this could increase the susceptibility of the cells and trigger their death. Therefore, it is necessary to consider the study of the pathogenesis of these neurodegenerative diseases so that the treatment can become effective [24,45].

On a positive note, hepatocytes, gastric cells, and bone marrow cells require fewer insertion sites for transcription factors, reducing the risk of tumorigenesis [16]. In addition, using bone marrow and pre-culture cells could be avoided to reduce research costs [16]. Stem cell reprogramming through iPSCs overcomes the immune rejection barrier, as they are autologous therapies [22]. Another point to mention is that partial cell reprogramming has emerged as a breakthrough for the in vitro and in vivo rejuvenation of different cell lines without completely losing the identity of the cell type, which opens the possibility of using it as a therapy to rejuvenate humans [65]; however, the understanding of the mechanism and risks is crucial for the safety of its implementation as a treatment [51].

Finally, a dilemma arises when using ESs or iPSCs. On the one hand, the use of ESs faces ethical concerns, since they come from the in vitro fertilization of human supernumerary embryos [66]. On the other hand, iPSCs come from patients’ somatic cells, thus presenting fewer ethical constraints and a low risk of immune rejection in vivo [30]. Several studies suggest that iPSCs might have a higher incidence of epigenetic and genetic aberrations than ESs due to the reprogramming process [67]. However, in recent years, more efficient and safer reprogramming techniques have been developed [68,69].

## 6. Conclusions

Yamanaka transcription factors have great potential for treating conditions such as cancer neurodegenerative diseases and for use in a rejuvenation treatment because they enable cell reprogramming. However, more research is needed to improve their safety and efficacy (Figure 4).

## Figures and Tables

**Figure 1 genes-14-01697-f001:**
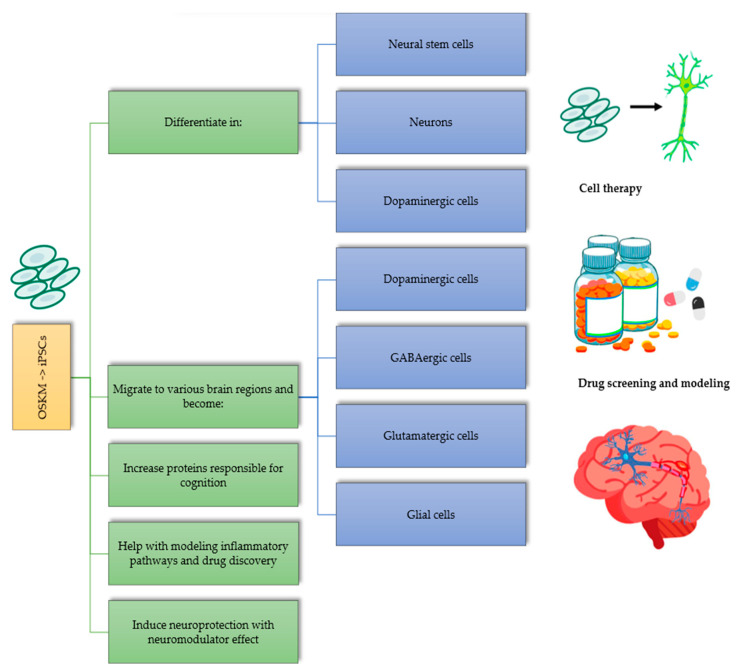
OSKM–iPSC positive outcomes in neurodegenerative diseases.

**Figure 2 genes-14-01697-f002:**
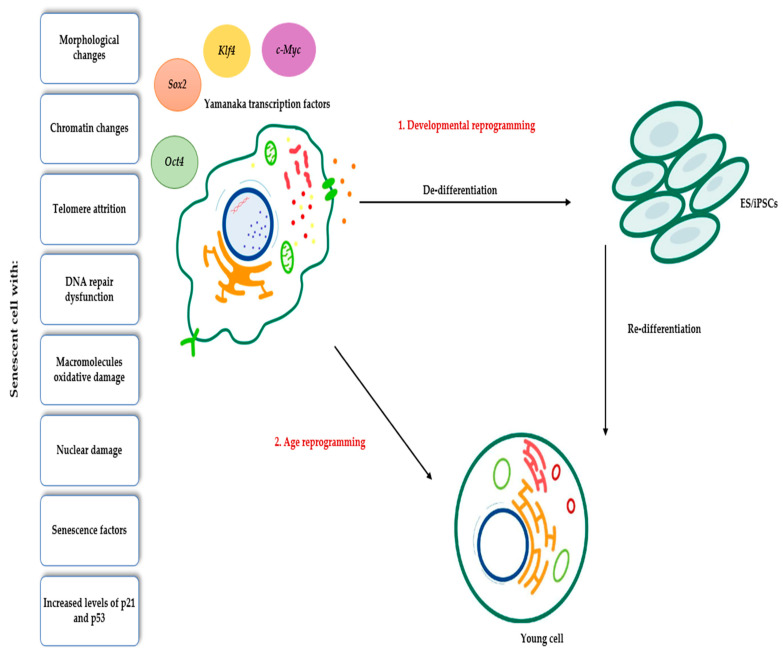
Characteristics of senescent cells and developmental and age reprogramming pathways.

**Figure 3 genes-14-01697-f003:**
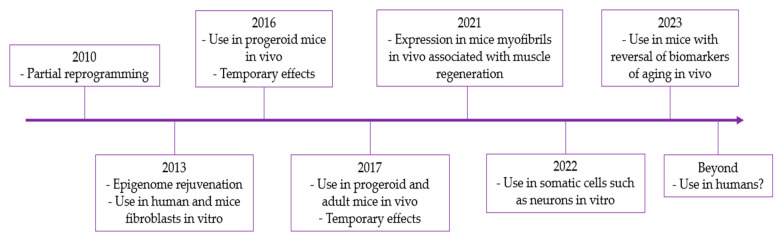
Timeline of the use of Yamanaka factors in rejuvenation.

**Figure 4 genes-14-01697-f004:**
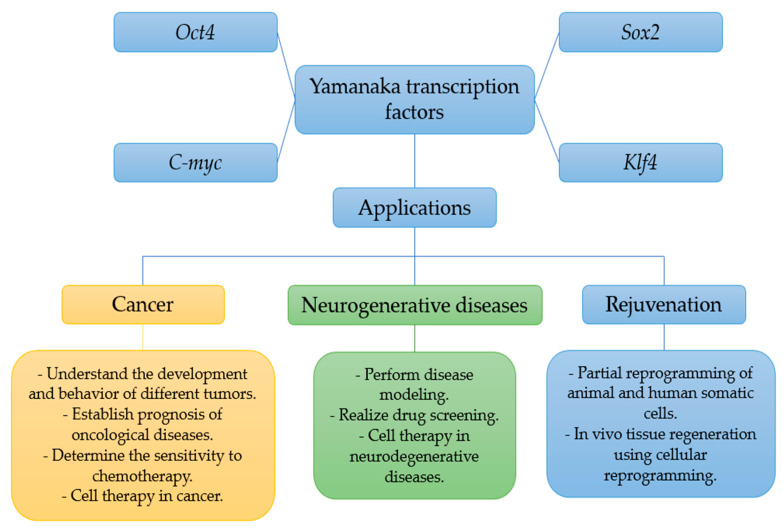
Summary of the different applications of the Yamanaka transcription factors.

## Data Availability

Not applicable.

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
