# Peer review of "Application of the Yamanaka Transcription Factors Oct4, Sox2, Klf4, and c-Myc from the Laboratory to the Clinic"

_genes, 2023, doi:10.3390/genes14091697_

Round 1
Reviewer 1 Report
The authors need to do major revision to the paper.
Most of their references are about 10 years and older. They need to incorporate more recent references, especially since this field-stem cell replacement therapy- and concerns associated with it are a hot topic.
They do not expand on a lot of their citations or explain it adequately. This leaves the reader quite confused. For example, "Results showed that Oct4-expressing colony-forming cells re-programmed to a primitive state in transgenic mice. Also, the research showed different characteristics of reprogramming pluripotent cells, such as time and best tissue." They need to expand on "primitive state" and "characteristics". Then they need to tie back every paper to the main context- for example, here how the primitive state and characteristics lead to cancer.
There is also a lot of research they haven't mentioned. For example, c-Myc is the major oncogenic factor in Wnt/B-catenin tumorigenicity, Oct4 in multiple cancers like ovarian, cervical, colorectal and liver, Nano is important in self-renewal of cd24+ cancer stem cells in hepatocellular cancer and Sox2 in at least 25 different cancers.
The authors also use grammatically wrong and difficult understand English. They would have to fix those.
Author Response
Point 1. Most of their references are about 10 years and older. They need to incorporate more recent references, especially since this field-stem cell replacement therapy- and concerns associated with it are a hot topic.
Response 1: we added recient references.
Point 2. They do not expand on a lot of their citations or explain it adequately. This leaves the reader quite confused. For example, "Results showed that Oct4-expressing colony-forming cells re- programmed to a primitive state in transgenic mice. Also, the research showed different characteristics of reprogramming pluripotent cells, such as time and best tissue." They need to expand on "primitive state" and "characteristics". Then they need to tie back every paper to the main context- for example, here how the primitive state and characteristics lead to cancer.
Response 2: we expand and explain our citations.
Point 3. There is also a lot of research they haven't mentioned. For example, c-Myc is the major oncogenic factor in Wnt/B-catenin tumorigenicity, Oct4 in multiple cancers like ovarian, cervical, colorectal and liver, Nano is important in self-renewal of cd24+ cancer stem cells in hepatocellular cancer and Sox2 in at least 25 different cancers.
Response 3: we added more information.
Point 4. The authors also use grammatically wrong and difficult understand English. They would have to fix those.
Response 4: we fixed our grammar and a native English speaker reviced it.

Reviewer 2 Report
1. Lane 12-14, OSKM allow reprogram somatic cells to induced pluripotent cells (iPSCs), and then used iPSCs to generate newly differentiated cells.
2. The conclusions in the the Introduction should cite the right references.
3. Introduction part, lack of one paragraph introduced the current understanding of somatic cell reprogramming by OSKM in the context of transcriptome/epigenetic (histone modification, chromatin accessibility, nucleosome reorganization, DNA methylation etc). And the application of iPSCs to differentiate to other cell types/ (Like NPCs, organ-like cells etc).
4. Lanes 50-51, this sentence is not correct described, should be modified.
5. Lanes 64-75. Methods section is not required for a review.
6. The format of references should follow the journal's guidance.
7. The reference is not enough for a review paper. More recent progress should be included in the review paper to make it more informative, like use urine cells to regenerate tooth like cells (Cai et al., Generation of tooth-like structures from integration-free human urine induced pluripotent stem cells, Cell Regeneration, 2013.).
8. Perspective:
Instead using OSKM, in lab research, scientists already developed small chemicals cocktails to induce somatic cells to ES/EPS (Like research led by Hongkui Deng lab), this kind of application should be also included in the discussion/perspective.
The authors should include more reference related to understanding of somatic cell reprogramming, Cited the following references, but not limit to.
1) Takahashi & Yamanaka. Induction of pluripotent stem cells from mouse embryonic and adult fibroblast cultures by defined factors. Cell, 2006.
2) Guan et al., Chemical reprogramming of human somatic cells to pluripotent stem cells. Nature, 2022.
Author Response
Point 1. Lane 12-14, OSKM allow reprogram somatic cells to induced pluripotent cells (iPSCs), and
then used iPSCs to generate newly differentiated cells.
Response 1: we changed lane 12-14.
Point 2. The conclusions in the the Introduction should cite the right references.
Response 2: we changed it.
Point 3. Introduction part, lack of one paragraph introduced the current understanding of somatic cell reprogramming by OSKM in the context of transcriptome/epigenetic (histone modification, chromatin accessibility, nucleosome reorganization, DNA methylation etc). And the application of iPSCs to differentiate to other cell types/ (Like NPCs, organ-like cells etc).
Response 3: We added information about it.
Point 4. Lanes 50-51, this sentence is not correct described, should be modified.
Response 4: We modified it.
Point5. Lanes 64-75. Methods section is not required for a review.
Response 5: we deleted methods section.
Point 6. The format of references should follow the journal's guidance.
Response 6: we follow the journal’s format.
Point 7. The reference is not enough for a review paper. More recent progress should be included in the review paper to make it more informative, like use urine cells to regenerate tooth like cells (Cai et al., Generation of tooth-like structures from integration-free human urine induced pluripotent stem cells, Cell Regeneration, 2013.).
Response 7: we added recient references.
Point 8. Perspective:
Instead using OSKM, in lab research, scientists already developed small chemicals cocktails to induce somatic cells to ES/EPS (Like research led by Hongkui Deng lab), this kind of application should be also included in the discussion/perspective.
The authors should include more reference related to understanding of somatic cell reprogramming, Cited the following references, but not limit to.
1) Takahashi & Yamanaka. Induction of pluripotent stem cells from mouse embryonic and adult fibroblast cultures by defined factors. Cell, 2006.
2) Guan et al., Chemical reprogramming of human somatic cells to pluripotent stem cells. Nature, 2022.
Response 8: we included it.

Round 2
Reviewer 2 Report
The revised manuscript has been extensively revised. It is now suitable for publication in Genes. Below are some comments before acceptance.
1. Nucleosome positioning during reprogramming was not included in the manuscript. Please include the content with the reference (PMID: 26639176).
2. Genes listed in the manuscript should be italic.
3. The resolution of the figures could be improved.